# Sol-Gel Processed TiO_2_ Nanotube Photoelectrodes for Dye-Sensitized Solar Cells with Enhanced Photovoltaic Performance

**DOI:** 10.3390/nano10020296

**Published:** 2020-02-10

**Authors:** Nikolai Tsvetkov, Liudmila Larina, Jeung Ku Kang, Oleg Shevaleevskiy

**Affiliations:** 1Solar Energy Conversion Laboratory, Institute of Biochemical Physics RAS, 119334 Moscow, Russia; larina@sky.chph.ras.ru; 2Graduate School of Energy, Environment, Water and Sustainability (EEWS), Department of Materials Science and Engineering, Korea Advanced Institute of Science and Technology (KAIST), 291 Daehak-ro, Yuseong-gu, Daejeon 34134, Korea; 3Department of Chemical Engineering and Applied Chemistry, Chungnam National University, Daejeon 34134, Korea

**Keywords:** dye-sensitized solar cells, nanotubes, charge traps, semiconductor-liquid interface, X-ray photoelectron spectroscopy

## Abstract

The performance of dye-sensitized solar cells (DSCs) critically depends on the efficiency of electron transport within the TiO_2_-dye-electrolyte interface. To improve the efficiency of the electron transfer the conventional structure of the working electrode (WE) based on TiO_2_ nanoparticles (NPs) was replaced with TiO_2_ nanotubes (NTs). Sol-gel method was used to prepare undoped and Nb-doped TiO_2_ NPs and TiO_2_ NTs. The crystallinity and morphology of the WEs were characterized using XRD, SEM and TEM techniques. XPS and PL measurements revealed a higher concentration of oxygen-related defects at the surface of NPs-based electrodes compared to that based on NTs. Replacement of the conventional NPs-based TiO_2_ WE with alternative led to a 15% increase in power conversion efficiency (PCE) of the DSCs. The effect is attributed to the more efficient transfer of charge carriers in the NTs-based electrodes due to lower defect concentration. The suggestion was confirmed experimentally by electrical impedance spectroscopy measurements when we observed the higher recombination resistance at the TiO_2_ NTs-electrolyte interface compared to that at the TiO_2_ NPs-electrolyte interface. Moreover, Nb-doping of the TiO_2_ structures yields an additional 14% PCE increase. The application of Nb-doped TiO_2_ NTs as photo-electrode enables the fabrication of a DSC with an efficiency of 8.1%, which is 35% higher than that of a cell using a TiO_2_ NPs. Finally, NTs-based DSCs have demonstrated a 65% increase in the PCE value, when light intensity was decreased from 1000 to 10 W/m^2^ making such kind device be promising alternative indoor PV applications when the intensity of incident light is low.

## 1. Introduction

Nanocrystalline dye-sensitized solar cells (DSCs) based on titanium dioxide (TiO_2_) nanostructured layers have been the subject of intensive studies over the last decades [1]. DSC is a class of green energy devices, functioning at molecular and nanoscale levels. Recently, the power conversion efficiency (PCE) of DSCs reached a confirmed record of 14.3% under standard illumination conditions AM1.5G [2]. It was also shown that unlike the conventional Si-based solar cells in which PCE gradually decreases with lowering the light intensity the DSCs demonstrate an opposite trend especially under ambient lighting where the efficiency approached 30% [3]. 

The charge recombination at the TiO_2_-dye-electrolyte interface is a principal limiting factor for increasing the efficiency of the DSCs. Therefore, the further improvement of the DSC performance under both high and low illumination conditions requires a detailed understanding of the charge transfer mechanisms across the TiO_2_-dye-electrolyte interface. A number of studies investigated the various aspects of the charge transfer processes in DSCs [3,4,5]. In DSCs, photon absorption by dye molecules initiates the creation of photo-excited electrons followed by the injection to the TiO_2_ conduction band from where the electrons are transferred toward the conducting electrode. Numerous studies have focused on matching the device absorption characteristics to the solar spectrum and enhancing the light collection ability of DSCs by the use of tandem systems, quantum dots and new types of dyes [6,7,8]. In spite of a lot of studies on the subject, the development of cell component materials and their modeling for optimization of optoelectronic structures for highly efficient DSCs remain hot research issues for current DSC technology. Among the critical components of a DSC, the photo-electrode has appealed to special attention from both scientific and technical viewpoints because of its important role in solar cell operation.

Mesoporous layers based on TiO_2_ nanoparticle (NPs) are generally used in DSCs as dye absorbers and electron transporting photoelectrodes [9,10]. However, the efficiency of the electron transfer across the photo-electrode toward the conducting electrode is limited by the surface trapping sites that cause the recombination and lower the carrier mobility [11,12]. The improvement of the charge transfer efficiency in the nanocrystalline TiO_2_ electrode may be gained by the replacement of the TiO_2_ nanostructures with alternative materials [13,14,15,16] or by doping of TiO_2_ with donor atoms like niobium (Nb) [9,10]. The effect of the doping on the Photovoltaics (PV) parameters are closely related to the trap states and the modification of the electronic structure of the TiO_2_ electrode. Given that Ti^4+^ 3d bands formed the CB band edge, the doping with Nb^5+^ influence the electronic structure of the TiO_2_-dye-electrolyte interface. It was shown that the Nb-doping of TiO_2_ electrodes in DSCs can improve the charge injection and transport by tailoring the electronic structure of the TiO_2_-dye-electrolyte interface [10]. The relationship between PV parameters level of Nb-doping was confirmed via XPS analysis of Fermi energy level shift and intraband defect states. The optimization of the doping level in the range of 0.5 to 3.0 mol% for the highest PCE yields the 2.5 mol% Nb [10]. Kim and coworkers obtained similar results, showing the enhancement of the electron injection efficiency upon the Nb-doping [17]. In a recent review on the subject, Roose et al. have analyzed the advantages and disadvantages of Nb-doping for improving the solar cell parameters [18]. The analysis of the research in the field led authors to the conclusion that tradeoff between the increase of the J_SC_ and V_OC_ decrease is required for improvement of the device PCE. The possible way to succeed is the synthesis of the low-defect TiO_2_ NPs along with the low-level of the Nb-doping [18].

In regards to the replacement of the TiO_2_ NP structure with alternative, TiO_2_ nanotubes (NTs) for DSCs offer several advantages due to increased light scattering, fast electron transportation and reduction of trap sites compared to TiO_2_ NPs [19,20]. Pristine and chemically modified TiO_2_ NTs have been studied as photoelectrodes for DSCs and energy storage systems [21,22]. To the best of our knowledge, the most of publications which deal with TiO_2_ NTs-based photoelectrodes report the fabrication of nanotubes by anodization of the Ti foils that is rather a complicated fabrication method [20,23,24,25]. Moreover, the reported efficiencies of the DSCs fabricated with TiO_2_ NTs produced by anodization do not exceed those for TiO_2_ NPs-based devices due to lower surface area of the TiO_2_ NTs which limits the amount of the absorbed dye molecules and decreases light-harvesting ability [20,26]. 

In this work, a sol-gel method [27,28] was used for the fabrication of pristine and Nb-doped TiO_2_ NPs and NTs. Pristine, Nb-doped TiO_2_ NPs and NTs were used for the fabrication of the layered photoelectrodes for DSCs. Scanning electron microscopy (SEM), transmission electron microscopy (TEM) and X–ray diffraction (XRD) were employed to characterize the morphology and crystal structure of the NPs and NTs. X-ray photoelectron spectroscopy and photoluminescence measurements revealed a higher concentration of oxygen-related defects at the surface of NPs-based electrodes compared to that based on TiO_2_ NTs which was attributed to the smaller amount of inter-particle connections in NTs-based electrode. The PCE obtained at AM1.5G conditions for the DSCs based on NTs was higher compared to that obtained for DSCs based on NPs. The observed improvement of the performance is attributed to the more efficient electron transport through the nanotube network that was confirmed by the electrochemical impedance spectroscopy (EIS) measurements. Moreover, TiO_2_ NTs-based DSCs have shown a better performance under low-light illumination conditions where they demonstrated a 65% increase of PCE value when light intensity was decreased from 1000 to 10 W/m^2^. 

## 2. Experimental

### 2.1. Synthesis of TiO_2_ Nanoparticles and Nanotubes 

Undoped and 2% Nb-doped TiO_2_ nanoparticles were obtained by the co-hydrolysis of Ti and Nb precursors. To prepare undoped TiO_2_ NPs, 3.2 mL of acetic acid was added to 20 mL of titanium isopropoxide (TTIP, 99.999%, Aldrich) and the mixture was stirred with a Teflon stir blade for 15 min at room temperature. Nb-doped TiO_2_ was synthesized by mixing TTIP with Nb precursor, niobium ethoxide (NbEth, 99.999%, Aldrich, St. Louis, MO, USA) according to the procedure described elsewhere [9]. TTIP to NbEth ratio was adjusted to obtained 2% Nb-doped TiO_2_. The mixture of metallic precursor and acetic acid was poured into 68.4 mL of deionized water under a stirring condition of about 800 rpm. Approximately 1 h of stirring was needed to complete the hydrolysis reaction and then 0.9 mL of 65% nitric acid was added to the solution. The solution temperature was increased up to 80 °C and was kept constant for 80 min under a refluxing condition with intensive stirring. The nanoparticles were grown hydrothermally by using the prepared colloidal solution in a 125 mL Teflon-lined mini-autoclave at 210 °C for 48 h. After the nanoparticle growth was completed, 0.52 mL of 65% nitric acid was added to the colloidal solution following by treatment with ultrasonic stirring for 15 min. The prepared mixture was washed three times with ethanol by centrifugation. 

Synthesized pristine and Nb-doped TiO_2_ NPs were used as a starting material for the synthesis of pristine and doped NTs. In the typical synthesis procedure, 12.5 g of TiO_2_ NPs were dissolved in 100 mL of 10M NaOH aqueous solution. The resulted solution was autoclaved for 12 h at 120 °C. Then 150 mL of 0.1 M HCl solution was added to the TiO_2_ solution. At the next step, the TiO_2_ NTs were obtained from the resulted mixture by centrifugation in water and ethanol.

### 2.2. Fabrication of DSCs

A fluorine-doped tin oxide (FTO)-coated glass (Solaronix, 8 Ω/□, Aubonne, Switzerland) was sequentially cleaned in acetone, in a 0.1 M HCl ethanolic solution, in a Triton X-100 aqueous solution, in water and in ethanol. After the cleaning, the FTO glass was treated with a 40 mM TiCl_4_ aqueous solution at 70 °C for 30 min to deposit a compact TiO_2_ blocking layer. A doctor-blade method was then used to coat the FTO glass with a TiO_2_ paste. The paste was prepared by mixing TiO_2_ NPs or NTs with a terpineol and ethyl cellulose ethanol solution. Finally, the coated layers were gradually sintered at 160 °C for 30 min, at 450 °C for 45 min and at 500 °C for 20 min and cooled down naturally. The thicknesses of TiO_2_ NPs- and NTs-based layers were around 10 μm.

For the dye coating, the photoelectrodes were dipped in a 0.3 mM of Cis-bis(isothiocyanato)bis(2,2-bipyridyl-4,4-dicarboxylato)ruthenium(II) (N3 dye) solution in a mixture of acetonitrile and tert-butyl alcohol (1:1 volume ratio) for 24 h. After the sensitizer uptake, the TiO_2_ photoelectrodes were washed with acetonitrile. 

To fabricate a platinum counter electrode the FTO glass was cleaned in the same way as the substrates for photoelectrodes and dried at 400 °C for 10 min to remove the traces of organic solvent. The FTO coated side of the glass substrate was moistened in the 5 mM H_2_PtCl_6_ (Aldrich, 99.9%) ethanolic solution and sintered at 400 °C for 20 min. The dye-sensitized TiO_2_ photo-electrode and Pt counter electrode were assembled into a sandwich-type cell and sealed with ionomer film (Surlyn 1702). The commercially available electrolyte AN-50 (Solaronix) was used to fulfill the fabrication of DSCs. The active area of the DSCs was varied from 0.14 to 0.18 cm^2^.

### 2.3. Characterization

The Nb content of the TiO_2_ particles was determined with the aid of X-ray photoelectron spectroscopy (XPS) measurements. The thickness of the film was measured with a Tencor Alpha Step (Buffalo, NY, USA)500 surface profile system. Scanning electron microscopy (SEM) measurements were provided using the Hitachi-S4800 microscope. Before taking the SEM measurements, TiO_2_ samples were coated with osmium using the ion sputter coater (HPC-1 SW, Vacuum Device Inc., Osaka, Japan). The crystal structure was determined with the aid of a Rigaku D/MAX-IIIC powder X-ray diffractometer (XRD) and a JEM-3010 (JEOL) transmission electron microscope. For the transmission electron microscopy (TEM) measurements, the sintered powder was scraped out from the FTO surface. The scraped powder was dissolved in ethanol and treated with ultrasonic stirring for 5 min. The resultant mixture was dropped on a TEM grid. UV-Vis absorption spectroscopy (JASCO spectrometer, model V-570) was used to compare the absorption spectra of the dye washed out from the TiO_2_ NPs- and NTs-based photoelectrodes. Low-temperature photoluminescence (PL) excitation was carried out using a He-Cd laser (Omnichrome, 2074) with a 325 nm excitation wavelength. EIS measurements were performed under open circuit conditions and 1000 W/m^2^ illumination in a frequency range of 10^−1^–10^6^ Hz with Ivium Compactstat potentiostat. The amplitude of the modulated voltage was 10 mV. The current density–voltage (J–V) characteristics of the DSCs fabricated in this study were measured under AM1.5G simulated illumination with a Spectra Physics Oriel 300W Solar Simulator. Neutral density filters in the illumination pass were used to measure the performance of the DSCs under low light intensities. A thermopile radiant power meter (Spectra Physics Oriel, model 70260) with a fused silica window was used to set the integrated intensity at 1000 W/m^2^. The integrated intensity was kept constant throughout the measurements by means of a digital exposure controller (Spectra Physics Oriel, model 68950). 

## 3. Results and Discussion 

### 3.1. Crystal Structure and Morphology Characterization

We have compared the crystal structure of pristine and Nb-doped TiO_2_ NP and NTs films. Figure 1a shows the XRD patterns of TiO_2_ NPs and NTs samples. In all the cases, the strong diffraction peak at 2θ = 25.5° with a (101) preferred orientation indicates the presence of the anatase TiO_2_ phase only. For Nb-doped samples, we have observed a slight peak shift toward lower 2-Theta values indicating the tensile strain of TiO_2_ lattice due to the incorporation of the Nb atoms.

The surface area of photo-electrode is an extremely important parameter that governs the amount of the dye loaded and consequently the light-harvesting ability. To compare the surface area of pristine and Nb-doped NPs- and NTs-based electrodes we performed the standard procedure of dye loading on electrodes of the same sizes and, then, washing out the dye with aqueous solutions of the same volume. Afterward, the optical absorption spectrum of each dye solution was recorded (Figure 1b). The typical absorption peaks of N3 dye were observed in all the spectra. We found that the peak intensities are slightly higher for dye solution desorbed from the TiO_2_-NPs sample compared to those of the TiO_2_-NTs sample. The result indicates the slightly higher surface area of the NPs-based electrode compared to that of the NTs-based electrode. At the same time, we observed that Nb doping does not affect the absorption spectra of desorbed dye pointing out that doping has a negligible effect on the surface area of the electrodes. We have also compared the optical properties of the NPs and NTs layers (Appendix A, Appendix A). We detected slightly higher absorption (Appendix A) of the NTs film which was due to slightly increased reflectance of NTs (Appendix A). The increased reflectance of NTs was attributed to the larger size of NTs compared to NPs.

We have investigated the electronic structure of the fabricated samples using XPS. The valence band spectra of pristine NPs and NTs are given in Figure 2a. The binding energy scale is referenced to the Fermi level. The position of the valence band maximum (VBM) was determined from the electron emission spectrum by means of a linear extrapolation of the onset of the valence band emission. The similar values of VBM about 2.8 below the Fermi level were found for NPs and NTs films. Considering the similar value of the bandgaps of 3.2 eV for both layers, the similar positions of conduction band maximum for the NPs- and NTs-based electrodes were deduced. Therefore, the comparable electron injection rates from the S* energetic level of the photo-excited dye molecules into the conduction band of the NPs and NTs photoelectrodes can be expected. We have also compared the valence band spectra on Nb-doped electrodes (Appendix A, Appendix A). For both Nb-doped electrodes, the VBM position was found to be located at around 3.0 eV being in line with previous reports [9]. Lower VBM position implies the lower position of conduction band minimum of TiO_2_ which in turn is favorable for enhancement of the electron injection from dye [9,29].

The energy distribution curves of photo-emitted electrons reflect the valence band density of states (Figure 2a). The electron emissions from O p_π_ (non-bonding) and O p_σ_ (bonding) orbitals are observed in a range of 3–10 eV [10,29]. The results of the deconvolution analysis of the valence band spectra for NPs and NTs films are shown in Figure 2a. The peaks centered at 5.2 eV (solid lines) are attributed to the electron emission from O p_π_ orbitals, while peaks at 7.4 eV (solid lines) are assigned to the electron emission from O p_σ_ orbitals. The ratios of the emission intensity of O p_π_ to O p_σ_ orbitals were found to be of around 1.5 and 2.3 for NPs- and NT-based layers, respectively. A similar behavior was demonstrated for Nb-doped electrodes (Appendix A, Appendix A).

The obtained results imply the presence of the larger amount of surface oxygen (non-bonded oxygen) at the surface of the NTs compared to that on NPs’ surface. The existence of the larger fraction of surface oxygen in the NTs can be the sign of the lower concentration of surface vacancies in this sample compared to NPs film [10]. It is known that surface oxygen vacancies and related Ti^3+^ state formation lead to the creation of electronic traps at the surface of TiO_2_ that promote the charge recombination with electrolyte redox species. The contact areas between NPs can be the origin of the structural defects within the TiO_2_ network [8]. We assume that those structural defects that can present in the junctions between TiO_2_ NPs or NTs can acts as the recombination centers during solar cell operation. The lower defect concentration in the NTs-based electrode is expected from surface morphology (Figure 3a,b). Indeed, the NTs are much longer compared to the NPs and thus within the electrode layers of the same thickness one can expect a smaller number of connections between TiO_2_ NTs compared to that between TiO_2_ NPs. Thus, the advantage of the TiO_2_ NTs layer structure over the layer structure of NPs can be utilized for alternative photoelectrodes in DSCs to decrease interface charge recombination. 

In order to further confirm the difference in the defect concentration in NPs and NTs films, we performed low-temperature PL measurements. Figure 2b shows the spectra of TiO_2_ NPs- and NTs-based layers taken at 10K. The PL spectra of both samples show a broad peak centered at around 540 nm. This peak is commonly attributed to the relaxation of self-trapped excitons [30] but also can be related to the photo-excited electron recombination on oxygen vacancy defects [31]. As can be seen in Figure 2b, the NPs-based sample shows much stronger PL intensity under UV light irradiation as compared to that of the NTs-based sample. The obtained result can assign to the faster recombination rate in NPs sample due to higher oxygen vacancy concentration at its surface [31]. Thus, based on the results of XPS and PL measurements we can conclude that the surface oxygen concentration on NPs film is higher compared to that on NTs’ surface. A similar behavior was observed for Nb-doped electrodes (Appendix A, Appendix A).

The representative SEM plane images of TiO_2_ NPs and NTs samples are shown in Figure 3a,b. TiO_2_-NPs sample demonstrates rhombic and cubic shaped anatase particles with a diameter of 10-35 nm. TiO_2_-NT sample consists of the long and thin nanotubes with a length of a few hundred nm and thickness of around 10 nm. Figure 3c shows the high resolution transmission electron microscopy (HRTEM) image of synthesized TiO_2_ NPs. The interplanar distance of the observed particle was found to be around 0.35 nm that corresponds to the lattice spacing of (101) anatase. The similar lattice spacing was found for synthesized nanotubes (Figure 3d). The results are in line with XRD analysis, where the strong diffraction peak at 2θ = 25.5° with a (101) preferred orientation indicated the presence of the anatase TiO_2_ phase only in both TiO_2_ NPs and NTs samples (Figure 1a). HRTEM image reveals that the thickness of the nanotube wall is around 2–3 nm and inside diameter of around 5 nm. This indicates that an internal diameter of nanotubes is large enough for the dye penetration into the inner space of nanotubes. Thus, the internal surface of nanotubes is available for dye absorption It is also should be noted that we did not see any significant difference in the morphology of pristine and Nb-doped electrodes (Figure 3 and Appendix A, Appendix A). Both pristine and Nb-doped layer demonstrate a similar particle size and shape.

### 3.2. Currents-Voltage Characteristics 

We have fabricated several sets of devices with pristine and Nb-doped TiO_2_ NPs and pristine and Nb-doped NTs. The PV parameters of 5–8 solar cells of each type were investigated. The current density-voltage (J-V) curves of best performing DSCs are shown in Figure 4 and Table 1 lists the values of the open-circuit voltage (V_OC_), the short-circuit current density (J_SC_), the fill factor (FF) and the PCE. We used both pristine and 2% Nb-doped NPs- and NTs-based layers as photoelectrodes since it is known that Nb doping in this concentration is favorable for improving charge transfer efficiency in the DSCs [8]. At the same time, we found that the Nb doping does not affect the morphology of NPs either the NTs morphology (Appendix A, Appendix A). 

The photovoltaic parameters and the performance of DSCs based on TiO_2_ NTs were found to be higher compared to those obtained for DSCs based on NPs for devices with both undoped and Nb-doped electrodes (Table 1, Figure 4). The application of Nb-doped TiO_2_ NTs as photo-electrode enables the fabrication of a DSC with an efficiency of 8.1%, which is 35% higher than that of a cell using a TiO_2_ NPs. First, of all, we observed that the J_SC_ obtained at AM1.5G conditions for the DSC based on TiO_2_ NTs was higher compared to those obtained for DSCs based on TiO_2_ NPs. Given that the active surface area of the NPs electrode is slightly higher compared to that of the NTs electrode (Figure 1b) the obtained improvement of the J_SC_ cannot be attributed to the higher light-harvesting at NTs electrode. Therefore, we can conclude that the J_SC_ increase is due to the more efficient electron transport through the nanotube network as it will be confirmed below.

Furthermore, the V_OC_ and FF were found to be higher for devices with TiO_2_ NTs photoelectrodes. These PV parameters depend on the recombination rate of the photo-injected electrons with the oxidized electrolyte species (I_3_^−^) which is the dominant reaction pathway in DSCs [32]:I_3_^−^ + 2e^−^ (TiO_2_) = 3I^−^(1)

The recombination rate, in turn, is related to the resistance at the TiO_2_/electrolyte interface. The correlation between resistance at the interface and V_OC_ of DSCs based on TiO_2_ electrodes with 2% doping levels will discuss below in terms of the EIS analysis. As mentioned above the interconnection between particles have structural defects that can trap electrons at the TiO_2_ surface leading to their recombination with electrolyte I_3_^−^ species (Equation (1)). The XPS results which revealed the lower oxygen vacancy concentration at the surface of the NTs electrode support the fact of the facilitation of electron transport within the NTs network due to the decreased amount of junctions within the electrode layer. The EIS analysis of the behavior of DSCs with pristine and Nb-doped NPs and NTs-based WE was performed in order to evaluate the proposed hypothesis. 

### 3.3. Impedance Spectroscopy Analysis

The investigation of the interfacial electron transfer for DSC with TiO_2_ NPs- and NTs-based photoelectrodes was carried out using EIS measurements under open-circuit conditions at AM1.5G illumination. Figure 5 shows the recorded impedance spectra (Nyquist plots) in the range of 10^−1^ to 10^5^ Hz and the equivalent circuit used for data fitting. 

All the DSCs exhibit three similar semicircles for the different frequency responses. Semicircles ω_1_, ω_2_ and ω_3_ are related to the charge transfer within the Pt counter electrode, the TiO_2_ layer and the electrolyte, respectively [33]. Two important parameters that reflect the efficiency of charge transfer within DSCs can be extracted from EIS spectra. Those are series resistance of the cell, R_S_, which reflects the resistance of the whole device and charge transfer resistance at the TiO_2_/electrolyte interface, R_W_, which in turn is related with the efficiency of the electron transfer across the TiO_2_ mesoporous network and recombination processes between of these elections and electrolyte redox species (Equation (1)) [9,22,33]. The R_S_ and R_W_ parameters of fabricated DSCs are summarized in Table 1. We have observed similar R_S_ values of DSCs with NPs and NTs electrodes that indicate the similar conductivity values of both photoelectrodes. At the same time, R_S_ values are decreasing upon doping indicating higher conductivity of Nb-doped photoelectrodes. 

The values of the charge-transfer resistance at the TiO_2_/electrolyte interface, R_W_, were estimated as a diameter of the middle-frequency ark in the Nyquist plots. This is a very important parameter that reflects the efficiency of the electron transfer across the TiO_2_ network that can primarily affect the V_OC_ and FF values [9,34,35]. At the same time, it should be noted that the other factors such as resistance at the Pt/FTO interface can also contribute to the middle frequency EIS response [33,36]. However, in the current study, all the cells were fabricated using the counter electrode prepared using the same methodology thus we are expecting the difference in the middle range response is due to the variation of the stricture of WE. For several samples under investigation, we observed that the R_W_ values for NPs-based DSCs were found to be around 20% smaller compared to those of NTs-based devices. At the same time, DSCs with NTs electrodes demonstrate higher V_OC_ and FF parameters. Thus, we confirmed a correlation between the R_W_ and V_OC_ and FF values. A high resistance at the TiO_2_/electrolyte interface implies the suppressed electron recombination with electrolyte redox species and consequently positively affect the PV parameters such as V_OC_ and FF. The increase of R_W_ in the cells employing TiO_2_ NTs is attributed to the lower concentration of oxygen vacancies at the surface of TiO_2_-NTs photo-electrode as was proven by analysis of the valence band spectra (Figure 2a) and PL measurements. Moreover, the structure of NTs-based photoelectrodes suggested fewer inter-surface junctions leading to a decrease in the electrical resistance of the TiO_2_ layer itself due to lower defect concentration. It can be the other factor that explains the increase in the J_SC_ of NTs-based solar cells [22].

### 3.4. DSCs operated under various light intensities

DSC is known to be a unique device compared to the other types of solar cells since that kind of device demonstrated an unusual trend when energy conversion efficiency is increased with the decrease in the light illumination intensity. This is important outcomes since it is making DSCs be perspective devices regarding indoor application when the intensity of incident light is relatively low. Here, we have investigated the effect of the light intensity on the performance of devices with NPs- and NTs-based photoelectrodes (Figure 6). DSCs exploiting NTs-based photoelectrodes were found to be more efficient at low-light illumination intensities compared to devices with NPs-based electrodes. Best performing cells demonstrated a 65% increase of PCE value when light intensity was decreased from 1000 to 10 W/m^2^. The efficiency surpasses the PCE of the DSCs of conventional art by 25%. 

## 4. Conclusions

In conclusion, pristine and Nb-doped TiO_2_ NPs and NTs have been synthesized by a sol-gel method and used for the preparation of mesoscopic photoelectrodes for DSCs. A comparative study of the structural and surface electronic properties of the prepared photoelectrodes was provided. XRD and HRTEM characterizations indicated the pure anatase phase for all the photo-electrodes. The XPS and PL analysis confirmed the higher surface oxygen concentration for TiO_2_ NPs than that for the TiO_2_ NTs indicating that morphology of NTs-based photoelectrode to be more attractive for utilization in DSCs due to lower surface defect concentration. The performances of TiO_2_ NTs-based DSCs at AM1.5G were found to be 10–15% higher than those obtained for TiO_2_ NPs-based DSCs. In TiO_2_ NTs-based DSCs the increased values of J_SC_, V_OC_ and FF were achieved. The obtained result ascribed to a high recombination resistance at the TiO_2_-NTs/electrolyte interface that suppresses the electron back transfer to the electrolyte. Moreover, NTs-based DSCs have shown a better performance under low-light illumination conditions where they demonstrated a 65% increase of PCE value when light intensity was decreased from 1000 to 10 W/m^2^. The improved performance of NTs-based DSCs under high and low illumination originates from the morphological advantages of a TiO_2_-NT network layer which able more efficient electron transport. Finally, we have shown that the Nb-doped TiO_2_-NT network has a large potential for use as a photo-electrode in high-performance DSCs and be promising alternative indoor Photovoltaics applications. 

## Figures and Tables

**Figure 1 nanomaterials-10-00296-f001:**
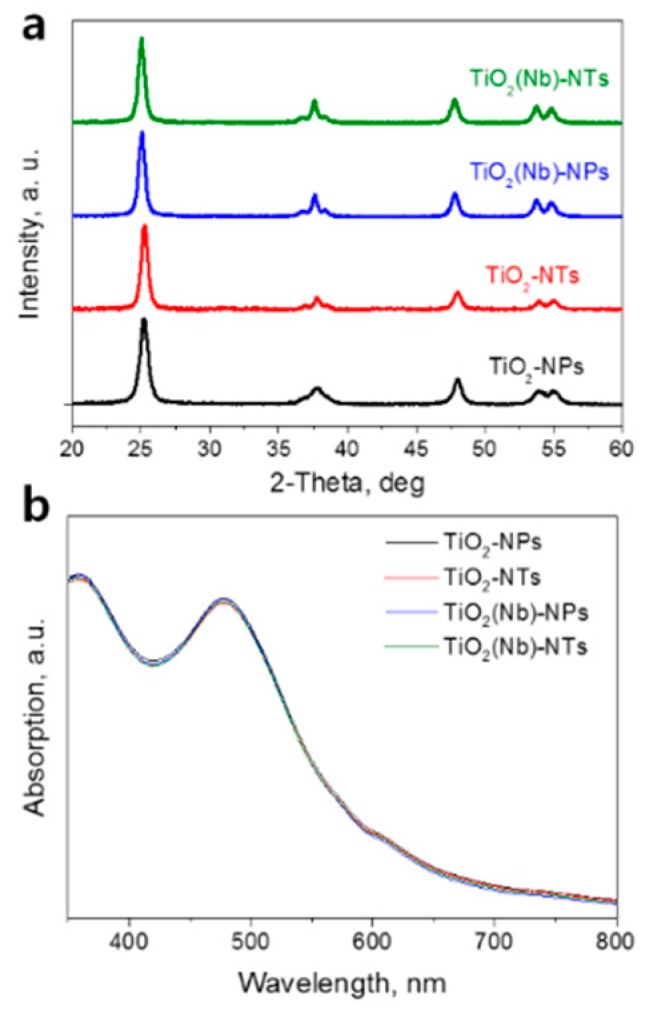
(**a**) X-ray diffraction (XRD) patterns of pristine and Nb-doped TiO_2_ nanoparticles (NPs)- and nanotubes (NTs)-based films. (**b**) Optical absorption spectra of N3 dye desorbed from the pristine and Nb-doped TiO_2_ NPs- and TiO_2_ NTs-based photoelectrodes.

**Figure 2 nanomaterials-10-00296-f002:**
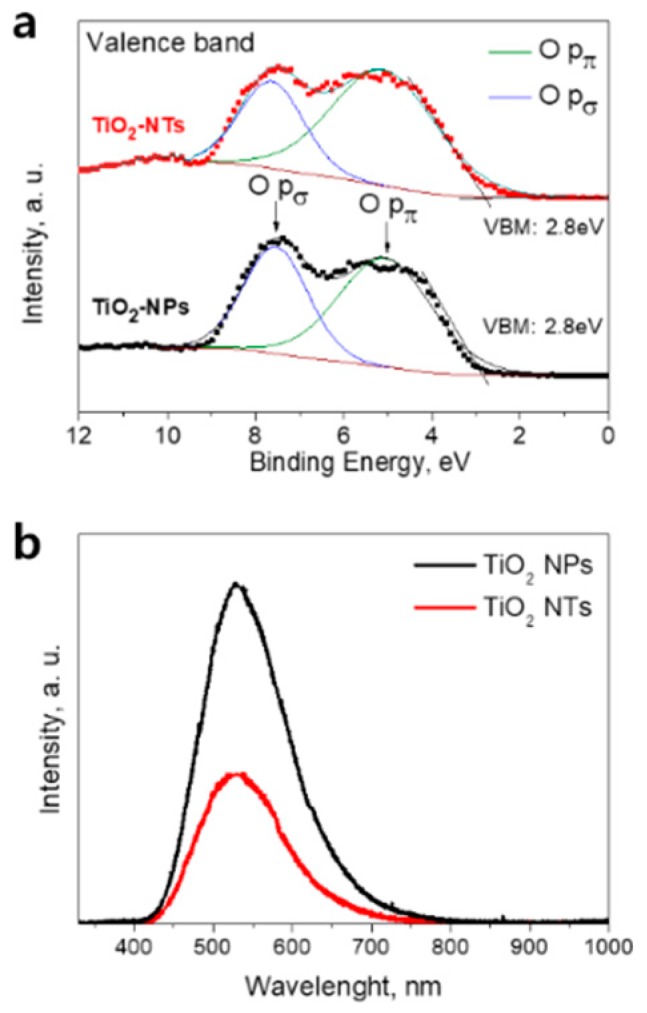
(**a**) Valence band X-ray photoelectron spectroscopy (XPS) spectra of pristine TiO_2_ NPs- and TiO_2_ NTs-based layers (symbols) and its Gaussian fit of XPS spectra (solid lines). (**b**) Photoluminescence (PL) spectra of NPs- and NTs-based TiO_2_ layers taken at 10 K.

**Figure 3 nanomaterials-10-00296-f003:**
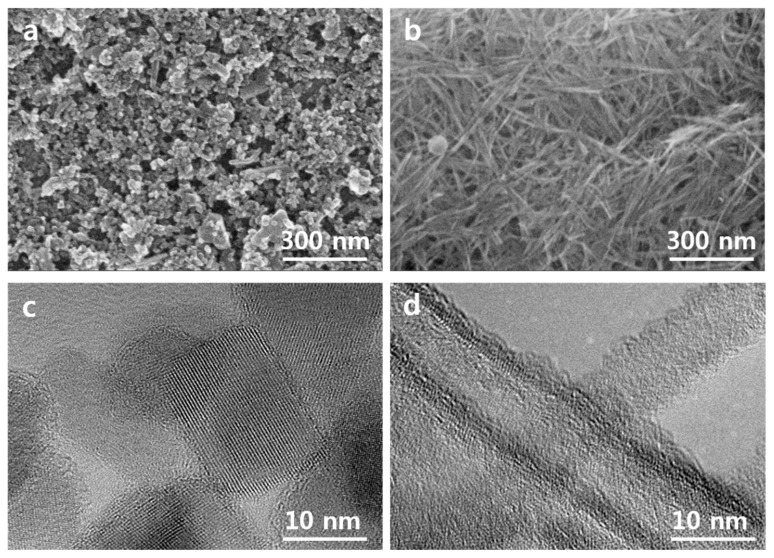
Scanning electron microscopy (SEM) micrographs of pristine (**a**) TiO_2_ NPs and (**b**) NTs. high resolution transmission electron microscopy (HRTEM) images of pristine (**c**) TiO_2_ NPs and (**d**) TiO_2_ NTs.

**Figure 4 nanomaterials-10-00296-f004:**
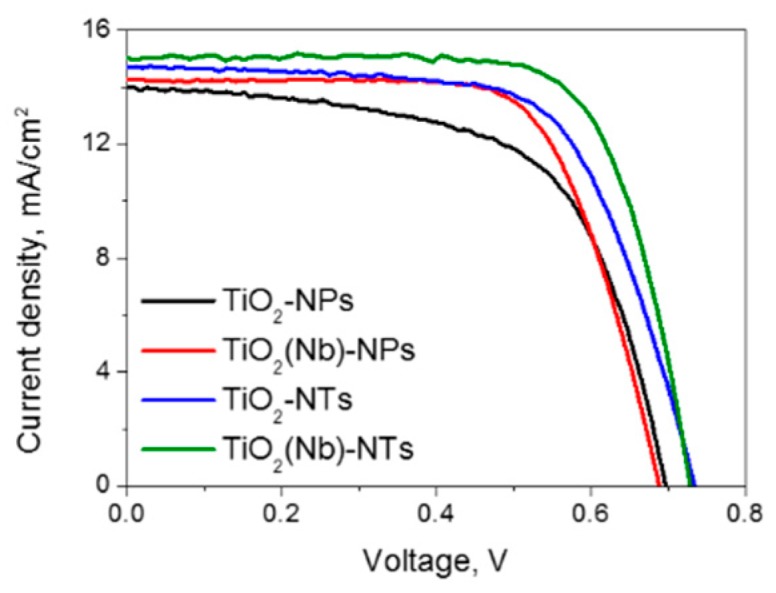
The current density-voltage curves of best-performing dye-sensitized solar cells (DSCs) fabricated with TiO_2_ NPs- and NTs-based photoelectrodes.

**Figure 5 nanomaterials-10-00296-f005:**
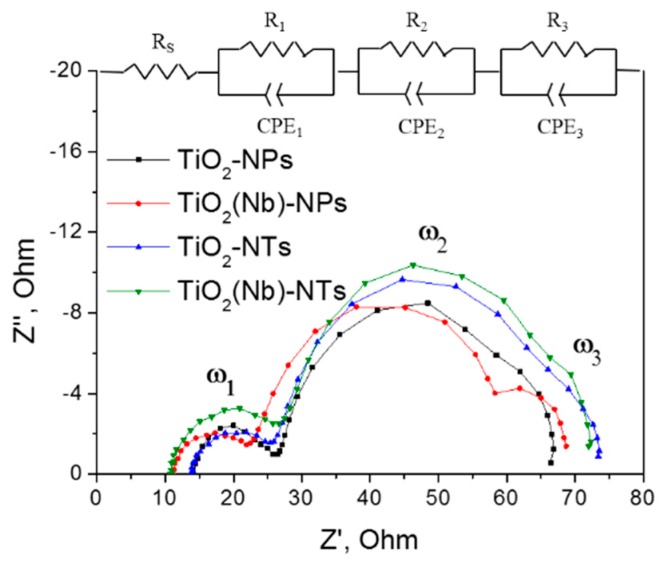
Nyquist plots of best performing DSCs based on TiO_2_ NPs- and NTs- photoelectrodes and equivalent circuits used for EIS data fitting.

**Figure 6 nanomaterials-10-00296-f006:**
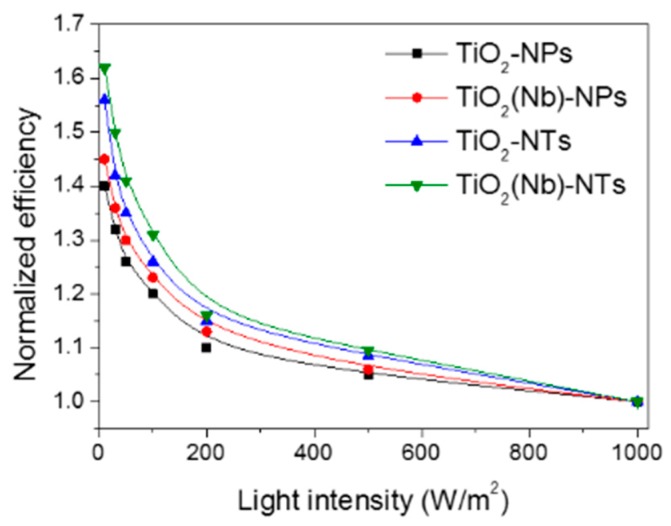
Normalized efficiency of pristine and Nb-doped TiO_2_ NPs- and NTs-based DSCs operated under various light intensities.

**Table 1 nanomaterials-10-00296-t001:** Photovoltaic and Impedance parameters of DSCs fabricated with TiO_2_ NPs- and NTs-based photoelectrodes (AM1.5G Illumination) and the series (R_S_) and TiO_2_/electrolyte interface resistance (R_W_) values extracted from EIS measurements. The parameters of best-performing devices are given in parentheses.

Photo-Electrode	V_OC_ (V)	J_SC_ (mA/cm^2^)	FF	PCE (%)	R_S_	R_W_
TiO_2_ NPs	0.69 ± 0.01 (0.70)	13.8 ± 0.2 (14.0)	0.59 ± 0.02 (0.61)	5.6 ± 0.4 (6.0)	14.5 ± 0.3 (14.2)	35.1 ± 2.4 (37.5)
Nb-doped TiO_2_ NPs	0.67 ± 0.01 (0.68)	14.2 ± 0.1 (14.3)	0.66 ± 0.01 (0.67)	6.3 ± 0.02 (6.5)	10.7 ± 0.4 (10.3)	36.6 ± 2.1 (38.7)
TiO_2_ NTs	0.72 ± 0.01 (0.73)	14.4 ± 0.2 (14.6)	0.64 ± 0.02 (0.66)	6.7 ± 0.4 (7.1)	14.6 ± 0.5 (14.1)	44.1 ± 1.5 (45.6)
Nb-doped TiO_2_ NTs	0.71 ± 0.01 (0.72)	14.8 ± 0.2 (15.0)	0.73 ± 0.02 (0.75)	7.7 ± 0.3 (8.1)	10.6 ± 0.2 (10.4)	42.9 ± 3.2 (46.1)

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
