# Peer review of "Sol-Gel Processed TiO_2_ Nanotube Photoelectrodes for Dye-Sensitized Solar Cells with Enhanced Photovoltaic Performance"

_nanomaterials, 2020, doi:10.3390/nano10020296_

Round 1
Reviewer 1 Report
This manuscript may be accepted after minor revision to the language and style
Author Response
No comments from reviewer
Reviewer 2 Report
In this work, the authors fabricate different DSSC devices by using TiO2 nanotubes photoelectrodes doped with Nb. In this sense, improved performance when compared with TiO2 nanoparticles is obtained, also in combination with the effect of the dopant. Although this approach is not entirely new, probably the preparation technique through a sol-gel method instead of the more extended anodization process is one of the points that can be highlighted, although for instance another reference on a parallel approach can be found: https://www.nature.com/articles/srep09305.
In general, the work is rather clear and the systematic comparative can be easily understood. Some questions and comments can be found below:
- Have been the TiO2 layers optically characterized? The fact of having different morphologies might somehow also influence the reflective/absorption properties.
-Regarding the XPS and the paragraph in lines 205-206, actually, the spectra shown in Figure S1 for the Nb-doped NPs and NTs seem quite similar. Were the emission intensity ratios calculated?
-Has the Ti 2p been analyzed by XPS?
Author Response
We are grateful for the reviewer’s thoughtful analysis of our work. In response to the reviewer’s comments and suggestions, we revised our manuscript and included additional experimental data.
We trust the referees will find that the revisions adequately address their suggestions.
Reviewer 2
In this work, the authors fabricate different DSSC devices by using TiO2 nanotubes photoelectrodes doped with Nb. In this sense, improved performance when compared with TiO2 nanoparticles is obtained, also in combination with the effect of the dopant. Although this approach is not entirely new, probably the preparation technique through a sol-gel method instead of the more extended anodization process is one of the points that can be highlighted, although for instance another reference on a parallel approach can be found: https://www.nature.com/articles/srep09305.
In general, the work is rather clear and the systematic comparative can be easily understood. Some questions and comments can be found below:
Q1. - Have been the TiO2 layers optically characterized? The fact of having different morphologies might somehow also influence the reflective/absorption properties.
Response to Q1. In response to the reviewer’s suggestions we performed the reflectance and absorption measurements for the NPs and NTs TiO2 samples. We found out that the reflectance of pressed powder NTs sample is slightly higher compared to that of NPs. The effect can be related to the increase of the reflectance of TiO2 nanostructures with the increase of the size [Yu et al., J. Mater. Chem., 2011,21, 532]. The following graphs were added to the Supporting Information and the corresponding discussion was included in the manuscript on the page 5.
“We have also compared the optical properties of the NPs and NTs layers (Figure S1, Supporting Information). We detected slightly higher absorption (Figure S1b) of the NTs film that was due to slightly increased reflectance of NTs (Figure S1a). The increased reflectance of NTs was attributed to the larger size of larger size of NTs compared to NPs.”
Finally, it should be noted that the absorption of TiO2 plays the minor role in DSCs operation since current generation is mostly taking place due to light absorption by dye molecules. Thus, we believe that the observed difference does not affect significantly the performance of fabricated DSCs.
Figure S1. (a) Reflectance and (b) absorption spectra of the NPs and NTs TiO2 photo-electrodes.
Q2. -Regarding the XPS and the paragraph in lines 205-206, actually, the spectra shown in Figure S1 for the Nb-doped NPs and NTs seem quite similar. Were the emission intensity ratios calculated?
Response to Q2. The valence band spectra of the Nb-doped samples (Figure S2) were very similar to those of non-doped samples. For Nb-doped samples, the higher intensity of O pσ peak was observed for NP samples. The calculated ratios of the emission intensity of O pπ to O pσ orbitals were found to be of around 1.7 and 2.1. For NPs- and NT-based layers, respectively. The results are provided in the Supporting Information (Table S1).
Q3. -Has the Ti 2p been analyzed by XPS?
Response to Q3. We have performed the XPS analysis of Ti 2p core-levels for the NPs and NTs TiO2. However, we did not observe any difference between the Ti 2p core-level spectrum taken from the NTs and NPs TiO2 samples. The representative Ti 2p core-level spectra and its deconvolution are given as Fig. R1 in attachment.
Reviewer 3 Report
The article by Tsvetkov et al. presents the synthesis of TiO2 nanotubes and the application in DSSC. The performances were improved with their design, the results are discussed with the help of EIS.
The article is well written, I noted only a few typos:
Line 40: “lowing” should be “lowering”
Line 51: “a lof of study” should be “a lot of studies”
Caption of figure 1: wavelength is misspelled.
“Figures 3c show” should be “Figures 3c shows”
This is certainly not exhaustive and a re-reading of the article will be beneficial to the latter.
Concerning the results, these are publishable in Nanomaterials in my opinion, provided that the following comments and questions are addressed:
why are there so many yellow highlights all over the article? Did the authors submit by mistake a work in progress draft?
Line 171: surface area discussion: I do not entirely agree with the link the authors make between surface area and dye loading. For instance, the size of the pores has to be taken into account as well. Usually, BET measurements allow to estimate the surface area of a material. Can such experiment be performed on bulk NP vs bulk NT? Regardless, the authors should not conclude on the surface area but only on the dye loading capacity of each material. Besides, the slight differences in absorption spectra of desorbed dyes are not significant, all samples seem to be equally loaded with dye molecules.
Line 195: the authors do not discuss the energy gap, asserting without evidence that the gap energy is the same between NP and NT materials: can they provide such evidence? Besides, what about the gap for Nb-doped materials vs pristine ones? A few comments are required here.
Line 219 “a smaller number of connections between TiO2 NTs”: one could imagine as well that less contact between particles leads to a less efficient percolation of the electrons. Can the authors comment on this?
Line 232: “The obtained result can assign to the faster recombination rate in NPs sample due to higher oxygen vacancy concentration at its surface”: I don’t understand how the authors state about the kinetics of the electron-hole recombination with just a steady state emission spectrum. If the spectra have been recorded in the same conditions for both NP and NT samples (same weight? Same volume? Have the measurements been done several times to check reproducibility?), one could conclude that there are more defect sites due to oxygen vacancies but this holds if one assumes that relaxation of self trapped excitons is the same for both samples, an assumption which should be verified.
Line 251: is there a reason why no HRTEM for Nb-doped NT is provided?
Table 1: since 5 to 8 cells were mounted for each test, please provide error bars.
Line 277: have the authors measured the dye loading? Or a UV spectrum of a photo-anode: this would be very interesting and would allow to highlight the impact of NT vs NP structure on the light harvesting efficiency.
A discussion comparing Nb doped vs undoped TiO2 in light of all EIS measurements is missing.
In conclusion, I recommend some major revisions before acceptance.
Author Response
We are grateful for the reviewer’s thoughtful analysis of our work. In response to the reviewer’s comments and suggestions, we revised our manuscript and included additional experimental data.
We trust the referees will find that the revisions adequately address their suggestions.
Response to comments of Reviewer #3
The article by Tsvetkov et al. presents the synthesis of TiO2 nanotubes and the application in DSSC. The performances were improved with their design, the results are discussed with the help of EIS.
Q1. The article is well written, I noted only a few typos:
Line 40: “lowing” should be “lowering”
Line 51: “a lof of study” should be “a lot of studies”
Caption of figure 1: wavelength is misspelled.
“Figures 3c show” should be “Figures 3c shows”
This is certainly not exhaustive and a re-reading of the article will be beneficial to the latter.
Response to Q1. We thank the reviewer for the provided detail revision of our manuscript. We revised the text of the manuscript. In the text of the revised manuscript, we fixed all typos mentioned above.
Line 57: “…issue … with “…issues..”Line 57: “…In the regards of…” with “…In regards to ...”
Line 213, 246, 262: “…The similar …” with “…A similar. .”
Line 322: “…very important …” with “…a very important...”
Line 330: “…At the same time…” with “…At the same time, …”
Line 341: “…unique…” with “…a unique …”
Line 330: “…At the same time… “with “…At the same time, …”
Line 330: “…At the same time…” with “…At the same time, …”
Q2. Why are there so many yellow highlights all over the article? Did the authors submit by mistake a work in progress draft?
Response to Q2. The yellow highlights were provided in the text since this version of the manuscript is the 2nd round of the revision. The yellow highlights s in the text are saved in response to an editor request.
Q3. Line 171: surface area discussion: I do not entirely agree with the link the authors make between surface area and dye loading. For instance, the size of the pores has to be taken into account as well. Usually, BET measurements allow us to estimate the surface area of a material. Can such an experiment be performed on bulk NP vs bulk NT? Regardless, the authors should not conclude on the surface area but only on the dye loading capacity of each material. Besides, the slight differences in absorption spectra of desorbed dyes are not significant, all samples seem to be equally loaded with dye molecules.
Response to Q3. We agree with the reviewer that the BET measurements can be a useful tool for detail characterization of the surface area of the samples. However, BET measurements deal with powder samples with the weight of a few grams thus the high enough sample surface area allows obtaining accurate BET results. During BET measurements the weight of the samples is comparing with and without absorbed gas. In case the low surface area of the sample, the amount of the absorbed gas is also low and the difference in the sample weight can be hardly detected. The samples in our study are thin films with a total TiO2 weight of micrograms. Our samples have much lower total surface area due to a small amount of material and the variations in the weight of the samples during gas absorption/desorption processes are very small, leading to the high uncertainty of the BET results. That is why we did not use the BET measurements in our particular case.
Q4. Line 195: the authors do not discuss the energy gap, asserting without evidence that the gap energy is the same between NP and NT materials: can they provide such evidence? Besides, what about the gap for Nb-doped materials vs pristine ones? A few comments are required here.
Response to Q4. We have included the absorption measurements of TiO2 NP and NTs in the Supporting Information as Fig. S1b. We have observed some increase in absorbance when NPs electrode was replaced with NTs (Fig. S1b). We ascribe this result to an increase in the reflectance of NT electrodes (Fig. S1a). Also, we did not observe any significant difference in the bandgap of doped NT and NP samples. Regarding the influence of the doping on the bandgap of TiO2, many studies already investigated this question, for example in Refs. 9,10 or 27 from the original submission.
Q5. Line 219 “a smaller number of connections between TiO2 NTs”: one could imagine as well that less contact between particles leads to a less efficient percolation of the electrons. Can the authors comment on this?
Response to Q5. We consider that the inter-connections between particles can have defects that can be the recombination centers (see for example Ref. 9). Given that NTs are much longer than NPs we assume that the number of interconnections in the NT electrodes is significantly smaller considering the same thickness of both photoelectrodes.
The additional comments regarding this point were included in the text of the revised manuscript.
Page 7, line “223-227”
“The contact areas between NPs can be the origin of the structural defects within the TiO2 network [8]. We assume that those structural defects that can present in the junctions between TiO2 NPs or NTs can acts as the recombination centers during solar cell operation. The lower defect concentration in the NTs-based electrode is expected from surface morphology (Figure 3 a,b).”
Q6. Line 232: “The obtained result can assign to the faster recombination rate in NPs sample due to higher oxygen vacancy concentration at its surface”: I don’t understand how the authors state about the kinetics of the electron-hole recombination with just a steady-state emission spectrum. If the spectra have been recorded in the same conditions for both NP and NT samples (same weight? Same volume? Have the measurements been done several times to check reproducibility?), one could conclude that there are more defect sites due to oxygen vacancies but this holds if one assumes that relaxation of self trapped excitons is the same for both samples, an assumption which should be verified.
Response to Q6.
It is well known that surface oxygen vacancies and related defect state formation lead to the creation of electronic traps at the surface of TiO2 that promote the charge recombination with electrolyte redox species. The existence of the larger fraction of surface oxygen in the sample is the sign of the lower concentration of surface vacancies. To compare the surface electronic structures of the samples under study, we applied the XPS analysis with extreme surface sensitivity (the depth of interest is less than 7 nm). For samples, we have collected 10 scans for VB and averaged data is given in Fig. 2. The ratio of the emission intensity of O pπ (non-bonding) to O pσ (bonding) orbitals were found to be 1.5 and 2.3 for NPs- and NT-based layers, respectively. Thus, the higher density of the surface oxygen vacancies and related defect states for the NP-based layer was proved by XPS results.
We also performed low-temperature PL measurements that allows us comparing relaxation rate of self-trapped excitons to confirm the difference in the defect concentration in NPs and NTs films. At the same time, we agree with the reviewer regarding the point that steady-state measurements as XPS cannot be used for direct measurement of electron-hole recombination of DSCs since those are surface sensitive and should be performed in extra high vacuum. However, XPS measurements are an extremely powerful tool for comparison of the electronic structure of semiconductors that in turn governs the charge transfer processes in photo-electrodes of DSCs [see for example Ref. 10].
Q7. Line 251: is there a reason why no HRTEM for Nb-doped NT is provided?
Response to Q6. TEM measurements of Nb-doped NT were not performed since we did not observe any difference between the morphology of doped and undoped samples in the SEM image. In our previous work, we deeply studied the effect of Nb-doping of the TiO2 electrode on the electronic structure TiO2 We have shown that doping does not affect the morphology of the samples.
[Nikolay T.; Larina, L.; Shevaleevskiy, O.; Ahn, B.T. Electronic structure study of lightly Nb-doped TiO2 electrode for DSCs. Energy Environ. Sci. 2011, 4, 1480. Tsvetkov, N.; Larina, L.; Shevaleevskiy, O.; Ahn, B.T. Effect of Nb Doping of TiO2 Electrode on Charge Transport in DSCs. J. Electrochem. Soc. 2011, 158, B1281.
Now the investigation of the effects of doping on the properties of the photo electrode is out of the scope of the current study.
We included Nb-doped electrodes in the current study to achieve better performing DSCs since the doping of the photo electrode is a well-known tool to improve the energy conversion efficiency of DSCs.
Q8. Table 1: since 5 to 8 cells were mounted for each test, please provide error bars.
Response to Q8. In response to the reviewer’s suggestions, we modified the Table 1 to reflect the deviations of the solar cells parameters.
Q9. Line 277: have the authors measured the dye loading? Or a UV spectrum of a photo-anode: this would be very interesting and would allow to highlight the impact of NT vs NP structure on the light-harvesting efficiency.
Response to Q9.
Yes, we estimated the amount of the dye loaded on electrodes using the standard procedure: the dye was loaded on the NP-based and NT-based electrodes of the same sizes and, then, washing out the dye with aqueous solutions of the same volume. Afterward, the optical absorption spectrum of each dye solution was recorded. The results are given in Figure 1b. The typical absorption peaks of N3 dye were observed in the spectra. There is no significant difference in peak intensity was observed. Thus we can expect the similar amount of the to be absorbed in the samples.
”Or a UV spectrum of a photo-anode: this would be very interesting and would allow to highlight the impact of NT vs NP structure on the light-harvesting efficiency.”
Commonly and particular, in our study, a fluorine-doped tin oxide (FTO)-coated glass was applied as a conductive substrate in DSCs. The working electrodes are prepared as follow: a compact TiO2 blocking layer (BL) is deposited on FTO and next the FTO/BL substrate is coated with a TiO2 paste. The state-of-the-art structure of DSCs suggested that the working electrode is illuminated from the FTO side.
FTO is a wide band gap semiconductor with the direct allowed transition and the direct band-gap value of around 3.4 eV. The transmittance of around 80% in the visible range. The analysis of the diffuse reflectance spectra for anatase and rutile TiO2 in terms of indirect optical absorption provided in our previous study yielded the Eg value of 3.2 eV for anatase. Thus the substrate materials are transparent in the visible region but can absorb efficiently the UV light and only minor part of UV light can reach NP or NT-based mp-electrode. Moreover the energy of this part of light is significantly smaller compare to the energy of visible light thus even there is any difference in the ability of UV light absorption between NP and NTs it will not affect the performance of the devices.
Q10. A discussion comparing Nb doped vs undoped TiO2 in light of all EIS measurements is missing.
Response to Q10. The influence of doping of the EIS behavior of DSCs was in detail investigated in the previous works [Nikolay T.; Larina, L.; Shevaleevskiy, O.; Ahn, B.T. Electronic structure study of lightly Nb-doped TiO2 electrode for DSCs. Energy Environ. Sci. 2011, 4, 1480. Tsvetkov, N.; Larina, L.; Shevaleevskiy, O.; Ahn, B.T. Effect of Nb Doping of TiO2 Electrode on Charge Transport in DSCs. J. Electrochem. Soc. 2011, 158, B1281].
In this study, we focused on the influence of working electrode morphology on the performance. Therefore, a well-known part of the discussion was excluded since it is not scientifically new. We would like to note that Nb-doped electrodes were included in this study to achieve better performing DSCs since doping is a well-known tool to improve the energy conversion efficiency of DSCs. The investigation of the effects of the doping on properties of photo electrode or DSC behavior is out of the scope of the current study.
Round 2
Reviewer 3 Report
All the comments I made have been addressed, I only notice that "b" is missing in figure 1. Regarding the UV-vis spectrum of a TiO2 layer I advised in my comments, I would like to react however: the authors are right to say that recording UV on a fully operational DSSC is a challenge, for all the reasons they mention. HOwever, one can easily prepare a single layer TiO2 on glass, dye it with N3 and then record the spectrum. THis can be found in many publications and provide many useful information on the dye monolayer. I nevertheless fully agree that dye loading in this case is probably enough for the discussion.
Summarily, I recommend this article for publication
Author Response
Response to Reviewer comment:
First, we greatly appreciate the Reviewer’s work on detail revision of our research and catching and clarify many issues particularly, with Fig.1.
We have fixed it.
Regarding the optical behavior of the TiO2 layer, we agree that this kind of study can be interesting and can be subject to further studies. However, here there is an additional challenge regarding the fabrication of single-layer TiO2 which will consist of NPs or what is even more difficult, NTs, since those are few hundred nm long.
Regarding NPs, in the area of perovskite solar cells, the procedure for deposition of layers of 100-200 nm is well developed, but the procedure cannot be applied to NTs which are of a few hundred nm length. For DSCs it not critical since the optimal thickness of the working electrode is of 10 microns and we can easily fabricate uniform layers of such thicknesses. With the decrease of the film thickness, the problem with uniformity of NPs or NTs-based layers can significantly affect the optical response.
This manuscript is a resubmission of an earlier submission. The following is a list of the peer review reports and author responses from that submission.
Round 1
Reviewer 1 Report
The relative literature is not given adequately.
Hydrothermal synthesis of Nb-doped TiO2 has been shown by X. Lu et al. Adv. Funct. Mater. 2010, p.509.
Nb-doped TiO2 nanotubes by anodization have been shown by M. Yang et al. Chem. Commun. 2011, p.2032
Hydrothermally prepared Nb-doped TiO2 tubular architectures have been synthesized by the Schmuki group (Electrochem. Commun, 2012, p.56). Also, see the work of Aleksandra Gardecka.
Also, in ref10, a previous study of the authors, the quantum efficiency for the devises containing Nb-doped nanoparticles was 8%. In this work, the corresponding efficiency is only 6.5%. Why is this difference? How many devices have the authors tested in order to get these efficiencies?
Also, the authors state that the NTs prepared by anodiazation show lower properties than the nanoparticles. It is not explained adequately why the NTs prepared by hydrothermal conditions are better than the nanoparticles as photoanode electrodes.
I can hardly see any novel information in this work
Reviewer 2 Report
Please see the attached file
